# EDINET-Bench:
# Evaluating LLMs on Complex Financial Tasks using Japanese Financial Statements

**Issa Sugiura**[1,2]**, Takashi Ishida**[1]**, Taro Makino**[1]**, Chieko Tazuke**[1]
**Takanori Nakagawa**[1]**, Kosuke Nakago**[1]**, David Ha**[1]
[1]Sakana AI, [2]Kyoto University

## Abstract

Large Language Models (LLMs) have made remarkable progress, surpassing human performance on several benchmarks in domains such as mathematics and coding. A key driver of this progress has been the development of benchmark datasets. In contrast, the financial domain poses higher entry barriers due to its demand for specialized expertise, and benchmarks remain relatively scarce compared to those in mathematics or coding. We introduce EDINET-Bench, an open-source Japanese financial benchmark designed to evaluate LLMs on challenging tasks such as accounting fraud detection, earnings forecasting, and industry classification. EDINET-Bench is constructed from ten years of annual reports filed by Japanese companies. These tasks require models to process entire annual reports and integrate information across multiple tables and textual sections, demanding expert-level reasoning that is challenging even for human professionals. Our experiments show that even state-of-the-art LLMs struggle in this domain, performing only marginally better than logistic regression in binary classification tasks such as fraud detection and earnings forecasting. Our results show that simply providing reports to LLMs in a straightforward setting is not enough. This highlights the need for benchmark frameworks that better reflect the environments in which financial professionals operate, with richer scaffolding such as realistic simulations and task-specific reasoning support to enable more effective problem solving. We make our dataset and code publicly available to support future research.

🤗 https://huggingface.co/datasets/SakanaAI/EDINET-Bench
⭕ https://github.com/SakanaAI/edinet2dataset
⭕ https://github.com/SakanaAI/EDINET-Bench

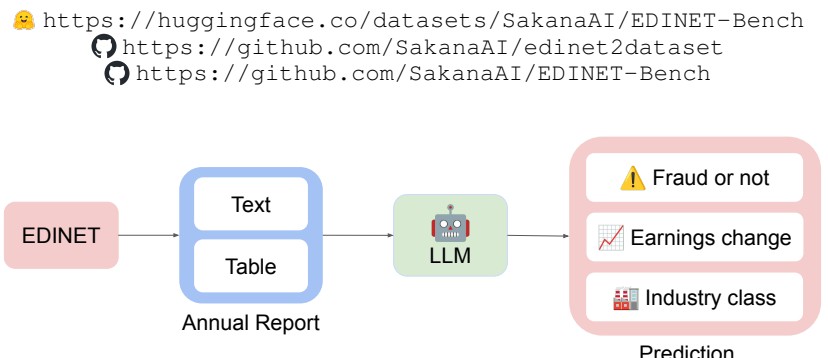

Figure 1: EDINET-Bench is a challenging benchmark evaluating LLMs on fraud detection, earnings forecasting, and industry classification from annual reports with text and tables.

## 1 Introduction

Large Language Models (LLMs) have recently demonstrated remarkable capabilities across diverse domains, including general knowledge, mathematics, and coding (OpenAI, 2024; Anthropic, 2024; Google, 2024; DeepSeek-AI, 2025b; Kimi Team, 2025). A key driver of this progress is the development of benchmark datasets. By evaluating models on benchmarks, researchers can identify

Table 1: Comparison of financial benchmarks. Expert-level reasoning refers to tasks that require synthesizing information across tables and text, including analyzing year-over-year trends, interpreting relationships between multiple financial tables, and combining these signals with narrative disclosures. Tasks such as fraud detection and earnings forecasting fall into this category, as they require comprehensive processing of both tabular and textual evidence together with domain-specific financial and accounting expertise.

| Benchmark | Input modality | Expert level reasoning | Language |
|---|---|---|---|
| FinQA (Chen et al., 2021) | Text, Table | | En |
| ConvFinQA (Chen et al., 2022b) | Text, Table | | En |
| FinanceBench (Islam et al., 2023) | Text | | En |
| FinanceMATH (Zhao et al., 2024) | Text, Table | | En |
| FAMMA (Xue et al., 2025) | Text, Image | | En |
| AuditBench (Wang et al., 2024) | Text, Table | | En |
| Japanese-lm-fin-harness (Hirano, 2024) | Text, Table | | Ja |
| EDINET-Bench (Ours) | Text, Table | ✓ | Ja |

current limitations, refine model architectures, training data, and methods, and ultimately climb the "benchmark hill". As a result, many existing benchmarks are now approaching saturation (OpenAI, 2025), motivating the creation of new benchmarks that are substantially more difficult and less prone to rapid saturation.

Prominent examples include Humanity's Last Exam (Phan et al., 2025), which consists of expert-level QA problems; SWE-Bench variants (Jimenez et al., 2024; Chowdhury et al., 2024; Deng et al., 2025), which require fixing real GitHub issues so that unit tests pass; and WebArena (Zhou et al., 2024), which evaluates agents operating real web browsers. Such benchmarks push models closer to real-world usability by mirroring expert-level tasks.

While benchmark research has advanced across many domains, progress in finance has lagged behind (Lee et al., 2025). Existing financial benchmarks primarily focus on relatively simple tasks, such as knowledge-based QA or data extraction from tables (Chen et al., 2021; 2022b; Islam et al., 2023). These are valuable for measuring basic domain knowledge but fall short of the kind of complex, high-stakes tasks that, if solved, would immediately translate to real-world utility.

To fill this gap, we introduce EDINET-Bench, an open-source financial benchmark constructed from ten years of real-world financial reports filed through Japan's Electronic Disclosure for Investors' NETwork (EDINET)[1] . EDINET-Bench presents demanding financial analysis challenges, including accounting fraud detection, earnings forecasting, and industry classification. These tasks require models to process complete annual reports, combine information from multiple tables and text sections, and demonstrate expert-level financial reasoning. To the best of our knowledge, this is the first open dataset and benchmark for accounting fraud detection.

We evaluate frontier LLMs on EDINET-Bench in zero-shot settings. Our results show that even state-of-the-art models struggle, performing only marginally better than logistic regression on binary classification tasks. This suggests that merely providing annual reports to LLMs in a simple setting is insufficient. Our findings underscore the need for benchmark frameworks that more closely mirror the environments in which financial professionals operate, incorporating richer scaffolding such as realistic simulations and task-specific reasoning support to facilitate more effective problem solving.

By releasing EDINET-Bench and its construction toolkit, we provide a foundation for studying expert-level financial reasoning with LLMs and for developing more powerful methods that can eventually support real-world financial applications.

## 2 BACKGROUND AND RELATED WORK

**Financial benchmarks.** A variety of financial benchmarks have been proposed in order to evaluate the financial knowledge and reasoning abilities of LLMs (Chen et al., 2021; Islam et al., 2023; Xie et al., 2024; Guo et al., 2025). For example, FinQA (Chen et al., 2021) and ConvFinQA (Chen

[1]https://disclosure2.edinet-fsa.go.jp/

et al., 2022b) are closed numerical reasoning question answering tasks related to financial analysis. Similarly, FinanceBench (Islam et al., 2023) is an open-book financial question answering benchmark dataset, and FinBen (Xie et al., 2024) is a benchmark-suit that contains 24 tasks. FAMMA (Xue et al., 2025) uses university textbooks and the CFA exam to construct a multimodal question answering benchmark. Several benchmarks exist in the Japanese financial domain (see Nakagawa et al. (2025) for a recent survey), including Japanese-lm-fin-harness (Hirano, 2024) and JCPA dataset (Masuda et al., 2023), which focus on Japanese financial exam questions. Recently, benchmarks such as FinanceMATH (Zhao et al., 2024), which evaluates an LLM's financial knowledge together with its mathematical reasoning, and AuditBench (Wang et al., 2024), which identifies subtle discrepancies between transaction data and financial statements, have been introduced. Despite these advances, existing benchmarks still center on tasks that can be solved with basic financial knowledge and relatively simple computations. As summarized in Table 1, most existing benchmarks rely on relatively simple QA settings that do not require expert-level reasoning. In contrast, EDINET-Bench poses a more complex challenge: the inputs consist of entire financial reports, and solving tasks such as accounting fraud detection requires integrating information across multiple tables and textual sections, demanding expert-level reasoning.

**Accounting fraud detection.** Accounting fraud detection involves identifying deliberate misstatements, manipulations, or discrepancies in financial reports. These reports are critical for numerous stakeholders, as investors base allocation decisions on them, while job seekers evaluate potential employers through their financial disclosures (Sou, 2018). Despite preliminary audits, fraudulent activities are frequently discovered only after publication, necessitating subsequent corrections (Japan Institute of Certified Public Accountants, 2024). Accounting fraud detection has been studied for many years, with pioneering work by Beneish (1999). After that, traditional machine learning approaches have been applied to accounting fraud detection (Perols, 2011; Dechow et al., 2011; Song et al., 2016; West & Bhattacharya, 2016; Kondo et al., 2019), but effectively processing textual information alongside numerical data in annual reports using LLMs remains underexplored.

**Earnings forecasting.** While earnings forecasting typically involves predicting the numeric magnitude of future earnings (Penman & Sougiannis, 1998; Monahan, 2018), a large body of work instead focuses on predicting the sign of the change in earnings (Ou & Penman, 1989; Chen et al., 2022a), which remains a difficult task for professionals. Kim et al. (2024) investigated if GPT-4, when provided with financial statements in which company names and fiscal years are anonymized, and without using any textual information, will outperform human analysts in predicting the direction of earnings for the following year. However, it is based on proprietary data and evaluation code were not made public. On the other hand, we will publish both the evaluation dataset and evaluation code to facilitate future research in this area.

**Industry prediction.** Industry prediction is a multi-class classification task that aims to predict the industry category based on security reports. While listed firms already are tagged with industry labels, e.g., by Securities Identification Code Committee (SICC), portfolio managers may want to adopt their own industry definitions, e.g., see Kimura & Nakagawa (2022) for a data-driven approach, or anticipate reclassifications as firms' business evolve or mergers reshape their operations. Evaluating whether LLMs can predict industries based on financial information such as balance sheets and profit/loss statements serves as an effective task for measuring LLMs' financial domain knowledge. Van Der Heijden (2022); Dolphin et al. (2023) used machine learning for this application.

**EDINET.** Electronic Disclosure for Investors' NETwork (EDINET) is a platform managed by the Financial Services Agency (FSA) of Japan that provides access to disclosure documents such as securities reports. This platform offers both a web interface and an EDINET API to access reports, allowing users to download annual, semi-annual, quarterly, and amended reports for the past ten years. In addition to PDFs, TSV files are also available, containing structured data where each row represents a record of individual attributes that exist in the annual report PDF. Detailed information for each disclosure item can be extracted and analyzed by parsing these attributes. EDINET is similar to EDGAR[2] in the United States. We use EDINET as the primary data source for our benchmark.

---

[2] https://www.sec.gov/edgar/search/

Table 2: Number of annual reports per fiscal year.

| Start of Fiscal Year | # Reports |
|---|---|
| 2014 | 3,638 |
| 2015 | 4,047 |
| 2016 | 4,065 |
| 2017 | 4,111 |
| 2018 | 4,120 |
| 2019 | 4,146 |
| 2020 | 4,206 |
| 2021 | 4,242 |
| 2022 | 4,263 |
| 2023 | 4,267 |
| 2024 | 586 |
| **Total** | **41,691** |

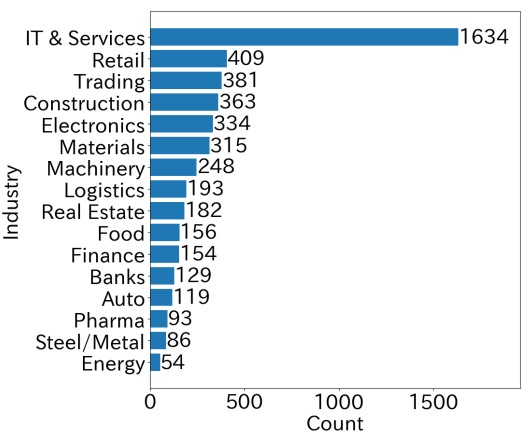

Figure 2: Number of companies per industry.

# 3 CONSTRUCTION OF EDINET-BENCH

In this section, we describe the construction method of EDINET-Bench. Overview of our construction pipeline is shown in Figure 1. Generally, EDINET-Bench is constructed by downloading Japanese listed companies' annual reports from EDINET and assigning labels for each task.

## 3.1 EDINET2DATASET

We first create edinet2dataset, a tool for downloading and parsing financial documents from EDINET. This tool primarily includes a download function for securities reports using the EDINET API and a parsing function to extract financial information from the downloaded report files in TSV format. The parsing process leverages Polars (Vink, 2021) to enable high-speed processing. edinet2dataset is inspired by the edgar-crawler from the Edger-Corpus (Loukas et al., 2021) and shares similar functionality. However, edinet2dataset is capable of extracting more detailed information, such as the current year's sales. The information extracted by edinet2dataset can be broadly categorized as follows: Meta: Metadata such as company name and EDINET code. Summary: Key financial indicators. BS: Balance sheet statement. PL: Profit and loss statement. CF: Cash flow statement. Text: Other textual information in the report, such as company history and business risk explanations.

## 3.2 EDINET-CORPUS

By using edinet2dataset, we collect all available annual reports and amended annual reports from EDINET for the period from April 2014 to April 2025, which includes approximately 40,000 documents (4,000 listed companies × 10 years). The number of annual reports obtained from EDINET over the past ten years for each year, as well as the number of companies per industry, are shown in Table 2 and Figure 2. We call this dataset the EDINET-Corpus, and it will be publicly available for reproducibility. The EDINET-Corpus serves as the primary data source for constructing EDINET-Bench, as described in the next section.

## 3.3 EDINET-BENCH

Using the aforementioned tool and corpus, we construct three challenging financial benchmark tasks: accounting fraud detection, earnings forecasting, and industry prediction.

**Accounting fraud detection.** We construct a binary classification dataset for accounting fraud detection. We collect both fraudulent and non-fraudulent reports in order to build this dataset. For the fraudulent reports, we first downloaded the past ten years of amended annual reports from EDINET, obtaining a total of 6,712 reports. An amended annual report is a document that discloses amendments to the original annual report. These amendments can be related to fraudulent activities or to non-fraudulent issues, such as misreporting the ownership percentage and share count of major

Table 3: Class distribution in accounting fraud detection task.

| Split | Non-fraud | Fraud | Total |
|-------|-----------|-------|-------|
| Train | 453 | 412 | 865 |
| Test | 102 | 122 | 224 |

Table 4: Class distribution in earnings forecasting task.

| Split | Decrease | Increase | Total |
|-------|----------|----------|-------|
| Train | 254 | 295 | 549 |
| Test | 157 | 294 | 451 |

Table 5: Class distribution in industry prediction task (test split).

| Industry | Count |
|----------|-------|
| Banking | 35 |
| Electronics & Precision Instruments | 34 |
| Automobiles & Transportation | 34 |
| Transportation & Logistics | 34 |
| Electricity, Gas & Energy Resources | 34 |
| Real Estate | 33 |
| Machinery | 33 |
| Steel & Nonferrous Metals | 33 |
| Materials & Chemicals | 32 |
| Finance (Excluding Banking) | 31 |
| Food | 31 |
| Construction & Materials | 29 |
| Trading Companies & Wholesale | 28 |
| Information & Communication Services | 28 |
| Pharmaceuticals | 24 |
| Retail | 23 |
| **Total** | **476** |

shareholders, under-reporting executive compensation, or failing to disclose critical information. To identify amendments linked to fraudulent activities, we extract text from amended annual reports in PDF format using pdfminer (Shinyama, 2016) , a specialized tool for PDF text extraction. Notably, amended reports explicitly state the reasons for corrections in text form, which provides direct evidence of irregularities. Leveraging this property, we analyze the extracted text with Claude 3.7 Sonnet, prompting the model to assess whether the stated reasons indicate fraud. This enables reliable and large-scale detection of fraudulent cases. The specific prompt used for this classification is detailed in Appendix B. As a result, 668 reports were identified as fraudulent and are labeled as fraudulent. We manually reviewed all cases labeled as fraudulent by reading the amendment reasons in the reports. Fewer than 5% of the fraud-labeled cases appeared unrelated (e.g., board member changes), suggesting a low error rate and supporting the reliability of the labels. For the non-fraudulent reports, we randomly sample 700 companies from all annual reports submitted over the past ten years, excluding those identified as fraudulent. From each selected company, we randomly choose one annual report from the reports of the company that have been downloaded. We then split the fraudulent and non-fraudulent samples into training and test sets, ensuring that data from the same company does not overlap between the two sets. This resulted in a dataset with 865 samples for training and 224 samples for testing, totaling 1,089 instances. Due to document parsing errors, the dataset was reduced to 534 fraudulent and 555 non-fraudulent samples, smaller than the initial 668 and 700 samples, respectively. The class distribution of the dataset is shown in Table 3.

**Earnings forecasting.**  We construct a binary classification dataset for earnings forecasting to predict whether a company's earnings will increase or decrease in the following fiscal year, given its current annual report[3]. To create this dataset, we randomly select 1,000 companies. For each company, we generate pairs of consecutive annual reports spanning two fiscal years, such as (2015, 2016), (2016, 2017), . . . , (2023, 2024). From these pairs, we randomly choose one pair for each company to include in the dataset (e.g., (2016, 2017)). For each sampled pair, we extract the value of "Profit Attributable to Owners of the Parent Company"[4] from the corresponding TSV files. We then compare the profit value between the two years: if the current year's profit exceeded the previous year's, the instance is labeled as an increase; otherwise, it is labeled as a decrease. Additionally, to evaluate baseline performance, we collect labels for earnings changes between two and one year prior as a naive baseline model. The dataset is split into train and test sets based on the fiscal year of the previous report in each pair. If the fiscal year started on or before January 1, 2020, it is included in

---

[3]Note that with our edinet2dataset, it is relatively easy to modify the task depending on the user's needs, e.g., change to a regression task or use other metrics.

[4]This value represents the portion of the consolidated net profit that is attributable to the shareholders of the parent company.

the train set; otherwise, it is assigned to the test set. This resulted in 549 samples for training and 451 for testing, totaling 1,000 instances. The class distribution for each split is shown in Table 4.

**Industry prediction.**   We construct a multi-class classification dataset for industry prediction to predict a company's industry based on its annual report. We use the industry information provided by EDINET for each company. The industry definitions follow the classification established by the SICC, which categorizes listed companies into 33 distinct industries. This classification, known as TOPIX-33, is widely recognized among Japanese investors. However, the granularity of 33 industries is considered too fine for our purposes. To enhance interpretability and simplify the prediction task, we consolidate similar industries into 16 broader categories[5]. This merging process is guided by the Japan Standard Industrial Classification published by the Ministry of Internal Affairs and Communications (MIC) and the SICC's official guidelines. We categorize all companies available in the EDINET-Corpus into 16 industry groups and randomly sample approximately 35 companies from each category. For each sampled company, the latest fiscal year's securities report is used as input for the industry prediction task. This process yields a test split of 496 samples with the class distribution shown in Table 5. In addition, we prepare a train split consisting of 900 examples.

## 4   EVALUATION

We evaluate the performance of current state-of-the-art LLMs in a zero-shot setting, and compare them to classical baselines trained on the train split.

### 4.1   EVALUATION SETUP

**Models.**   We evaluate several LLMs in a zero-shot setting.   For closed-source models, we include GPT-4o (`gpt-4o-2024-11-20`), o4-mini (`o4-mini-2025-04-16`), GPT-5 (`gpt-5-2025-08-07`), Claude 3.5 Haiku (`claude-3-5-haiku-20241022`), Claude 3.5 Sonnet (`claude-3-5-sonnet-20241022`), and Claude 3.7 Sonnet (`claude-3-7-sonnet-20250219`) (OpenAI, 2024; Anthropic, 2024).   For open-source models, we evaluate Kimi-K2, DeepSeek-V3, DeepSeek-R1, and Llama 3.3 70B (Kimi Team, 2025; DeepSeek-AI, 2025b;a; Llama Team, 2024). In addition, we evaluate a Llama-3.2-1B model fine-tuned on the train split. For classical models, we use Logistic Regression, Random Forest, and XGBoost (Cox, 1958; Breiman, 2001; Chen & Guestrin, 2016).

**Sampling parameters.**   For inference, the temperature parameter is set to 0.0 and the maximum token length for generation is set to 4096. For GPT-4o, the random seed is fixed at 0.

**Prompt.**   We use the system prompt "You are a financial analyst" across all tasks.  For binary classification tasks (accounting fraud detection and earnings forecasting), models are prompted to output: (1) a prediction (0 or 1), (2) a probability value (0 to 1), and (3) reasoning based on securities report information. For multi-class industry prediction, models output the predicted industry class and classification reasoning. The fraud detection prompt is shown in Figure 3, with other task prompts provided in Appendix A.

**Input formats.**   We extract the following items from the reports for prediction: Balance Sheet (BS), Cash Flow (CF), Profit and Loss (PL), Consolidated Summary (Summary), and Text Block (Text). These items are concatenated, appended to the end of the instruction prompt, and fed into the models in a zero-shot manner. To assess how information quantity affects LLM performance, we tested three configurations: Summary only, BS+CF+PL+Summary, and all items (BS+CF+PL+Summary+Text). Note that BS, CF, PL, and Summary consist of tabular data and do not include company names or exact year information, whereas Text contains unstructured textual information.

**Cost and processing time analysis.**   Each annual report in EDINET-Bench contains approximately 30,000 tokens spanning all sections. The average output length is around 500 tokens. For Claude 3.7

---

[5]In TOPIX-17, the number of cases in the industries of "Energy Resources" and "Electricity & Gas," which are loosely related domains, was smaller compared to other industries. Therefore, these two were merged to create 16 industries.

---

**Prompt for accounting fraud detection**

Please analyze the following information extracted from a Japanese company's securities report for any signs of fraudulent activities. Please note that some data may be missing and represented as "-" due to parsing errors. The report has been verified by a certified public accountant, and the numerical values are consistent and correct from a calculation perspective. Therefore, please focus your analysis on non-numerical inconsistencies or logical red flags that could suggest fraud. Respond in the following format: JSON: "'json { "reasoning": "string", "prob": float (between 0 and 1, probability that the report is fraudulent), "prediction": int (0: No fraud, 1: fraud) } "' The extracted securities report is as follows:

---

Figure 3: Prompt for accounting fraud detection.

Sonnet, which costs $3 per million tokens for input and $15 per million tokens for output, the cost per report is approximately $0.1. Regarding processing time, Claude 3.7 Sonnet can handle each case in about 10 seconds.

**Evaluation metrics.**    For accounting fraud detection and earnings forecasting, we use ROC-AUC (Receiver Operating Characteristic Area Under the Curve) and MCC (Matthews Correlation Coefficient) as evaluation metrics, and for industry prediction, we use accuracy.

### 4.2    RESULTS

Table 6 shows the performance of each model on each task in EDINET-Bench. In the following, we discuss the results for each task.

**Accounting fraud detection.**    In the accounting fraud detection task, the random forest achieves the highest performance. All LLMs struggle, with even the state-of-the-art Claude 3.5 Sonnet only marginally outperforming logistic regression in terms of ROC-AUC. Nevertheless, most models benefit substantially from incorporating textual information, suggesting that narrative disclosures provide useful signals for fraud detection. In some cases, model reasoning explicitly referred to auditor reputation (e.g., "Their financial statements have been audited by a reputable firm, Azusa Audit Corporation") as part of the decision-making process, highlighting a potential bias towards well-known auditing corporations. We further examine the feature importance of the logistic regression model, as presented in Table 7. This reveals that total comprehensive income and total assets are influential factors in non-fraud predictions, suggesting a bias towards predicting larger companies as non-fraud and smaller ones as fraud.

**Earnings forecasting.**    For the earnings forecasting task, many LLMs struggle to deliver precise predictions. Even GPT-5 achieves only a ROC-AUC of 0.65, aligning with the low forecasting performance reported by Xie et al. (2024). This limitation likely stems from the challenge of predicting next-year earnings based solely on the current securities reports. Unlike the fraud detection task, incorporating text information do not enhance performance in earnings forecasting, suggesting a weaker reliance on textual factors such as company size and audit firm reputation.

**Industry prediction.**    In the industry prediction task, all models demonstrate predictive performance significantly higher than random guessing. Furthermore, across all models, expanding the input from Summary to Summary+BS+CF+PL results in improved performance. Notably, Claude 3.5 Sonnet achieves state-of-the-art performance with an accuracy score of 0.41 when provided with Summary+BS+CF+PL. These results indicate that industry prediction is a relatively simpler task compared to accounting fraud detection and earnings forecasting, with predictions often possible using only securities reports. This likely reflects differences in asset composition, such as varying proportions of cash, loans, and securities across industries.

Table 6: Performance of each model on each task in EDINET-Bench. For LLMs, the scores represent the mean $\pm$ standard deviation over three runs. **Bold** indicates the best score. For industry prediction, the score for input formats that include the text section is omitted, as the text section of annual reports often explicitly states the company's industry, and such cases are indicated as "–". [†] denotes models that are trained on the train split.

| Model | Input Setup | Fraud Detection | | Earnings Forecasting | | Industry Prediction |
|---|---|---|---|---|---|---|
| | | ROC-AUC ↑ | MCC ↑ | ROC-AUC ↑ | MCC ↑ | Accuracy ↑ |
| **Closed-source models** | | | | | | |
| Claude 3.5 Haiku | Summary | $0.61 \pm 0.01$ | $0.19 \pm 0.02$ | $0.41 \pm 0.01$ | $-0.02 \pm 0.02$ | $0.09 \pm 0.00$ |
| | Summary+BS+CF+PL | $0.60 \pm 0.01$ | $0.18 \pm 0.03$ | $0.45 \pm 0.00$ | $0.00 \pm 0.05$ | $0.13 \pm 0.00$ |
| | Summary+BS+CF+PL+Text | $0.67 \pm 0.00$ | $0.28 \pm 0.02$ | $0.44 \pm 0.01$ | $-0.02 \pm 0.01$ | – |
| Claude 3.5 Sonnet | Summary | $0.64 \pm 0.01$ | $0.05 \pm 0.03$ | $0.54 \pm 0.01$ | $0.08 \pm 0.01$ | $0.24 \pm 0.01$ |
| | Summary+BS+CF+PL | $0.63 \pm 0.03$ | $0.18 \pm 0.03$ | $0.55 \pm 0.01$ | $0.10 \pm 0.02$ | $\mathbf{0.41 \pm 0.00}$ |
| | Summary+BS+CF+PL+Text | $0.73 \pm 0.02$ | $0.32 \pm 0.02$ | $0.52 \pm 0.02$ | $0.08 \pm 0.02$ | – |
| Claude 3.7 Sonnet | Summary | $0.59 \pm 0.01$ | $0.10 \pm 0.02$ | $0.55 \pm 0.01$ | $0.06 \pm 0.02$ | $0.24 \pm 0.01$ |
| | Summary+BS+CF+PL | $0.58 \pm 0.02$ | $0.09 \pm 0.04$ | $0.58 \pm 0.01$ | $0.13 \pm 0.01$ | $0.39 \pm 0.01$ |
| | Summary+BS+CF+PL+Text | $0.67 \pm 0.01$ | $0.25 \pm 0.02$ | $0.61 \pm 0.01$ | $0.16 \pm 0.02$ | – |
| GPT-4o | Summary | $0.59 \pm 0.00$ | $0.16 \pm 0.02$ | $0.40 \pm 0.00$ | $-0.17 \pm 0.00$ | $0.14 \pm 0.01$ |
| | Summary+BS+CF+PL | $0.61 \pm 0.01$ | $0.19 \pm 0.02$ | $0.41 \pm 0.00$ | $-0.13 \pm 0.01$ | $0.19 \pm 0.01$ |
| | Summary+BS+CF+PL+Text | $0.69 \pm 0.01$ | $0.29 \pm 0.02$ | $0.41 \pm 0.00$ | $-0.14 \pm 0.00$ | – |
| o4-mini | Summary | $0.53 \pm 0.00$ | $0.01 \pm 0.08$ | $0.51 \pm 0.00$ | $0.05 \pm 0.02$ | $0.19 \pm 0.01$ |
| | Summary+BS+CF+PL | $0.52 \pm 0.01$ | $0.04 \pm 0.05$ | $0.54 \pm 0.03$ | $0.13 \pm 0.04$ | $0.27 \pm 0.01$ |
| | Summary+BS+CF+PL+Text | $0.61 \pm 0.01$ | $0.10 \pm 0.05$ | $0.58 \pm 0.01$ | $0.15 \pm 0.01$ | – |
| GPT-5 | Summary | $0.56 \pm 0.00$ | $0.00 \pm 0.00$ | $0.58 \pm 0.00$ | $0.09 \pm 0.02$ | $0.21 \pm 0.01$ |
| | Summary+BS+CF+PL | $0.62 \pm 0.03$ | $0.08 \pm 0.03$ | $0.62 \pm 0.01$ | $0.20 \pm 0.02$ | $0.31 \pm 0.01$ |
| | Summary+BS+CF+PL+Text | $0.67 \pm 0.01$ | $0.07 \pm 0.04$ | $\mathbf{0.65 \pm 0.01}$ | $\mathbf{0.21 \pm 0.01}$ | – |
| **Open-weight models** | | | | | | |
| Kimi-K2 | Summary | $0.58 \pm 0.02$ | $0.10 \pm 0.08$ | $0.46 \pm 0.00$ | $-0.06 \pm 0.00$ | $0.14 \pm 0.01$ |
| | Summary+BS+CF+PL | $0.54 \pm 0.02$ | $0.08 \pm 0.03$ | $0.53 \pm 0.00$ | $0.04 \pm 0.00$ | $0.26 \pm 0.01$ |
| | Summary+BS+CF+PL+Text | $0.59 \pm 0.00$ | $0.11 \pm 0.05$ | $0.55 \pm 0.02$ | $0.07 \pm 0.07$ | – |
| DeepSeek-V3 | Summary | $0.61 \pm 0.03$ | $0.21 \pm 0.02$ | $0.39 \pm 0.01$ | $-0.14 \pm 0.02$ | $0.12 \pm 0.00$ |
| | Summary+BS+CF+PL | $0.59 \pm 0.02$ | $0.12 \pm 0.04$ | $0.40 \pm 0.01$ | $-0.13 \pm 0.01$ | $0.15 \pm 0.01$ |
| | Summary+BS+CF+PL+Text | $0.55 \pm 0.03$ | $0.07 \pm 0.06$ | $0.39 \pm 0.01$ | $-0.15 \pm 0.02$ | – |
| DeepSeek-R1 | Summary | $0.54 \pm 0.04$ | $0.01 \pm 0.08$ | $0.43 \pm 0.01$ | $-0.06 \pm 0.01$ | $0.11 \pm 0.02$ |
| | Summary+BS+CF+PL | $0.56 \pm 0.01$ | $0.09 \pm 0.02$ | $0.46 \pm 0.00$ | $-0.01 \pm 0.01$ | $0.16 \pm 0.01$ |
| | Summary+BS+CF+PL+Text | $0.63 \pm 0.01$ | $0.15 \pm 0.04$ | $0.44 \pm 0.01$ | $-0.05 \pm 0.02$ | – |
| Llama 3.3 70B | Summary | $0.58 \pm 0.01$ | $0.11 \pm 0.01$ | $0.42 \pm 0.01$ | $-0.09 \pm 0.03$ | $0.10 \pm 0.01$ |
| | Summary+BS+CF+PL | $0.59 \pm 0.00$ | $0.11 \pm 0.01$ | $0.41 \pm 0.01$ | $-0.11 \pm 0.02$ | $0.14 \pm 0.01$ |
| | Summary+BS+CF+PL+Text | $0.56 \pm 0.04$ | $0.06 \pm 0.06$ | $0.42 \pm 0.01$ | $-0.04 \pm 0.01$ | – |
| Llama-3.2-1B SFT[†] | Summary | $0.61 \pm 0.01$ | $0.18 \pm 0.05$ | $0.52 \pm 0.01$ | $0.03 \pm 0.03$ | $0.25 \pm 0.01$ |
| | Summary+BS+CF+PL | $0.66 \pm 0.02$ | $0.17 \pm 0.01$ | $0.52 \pm 0.01$ | $0.04 \pm 0.02$ | $0.26 \pm 0.02$ |
| | Summary+BS+CF+PL+Text | $0.54 \pm 0.02$ | $0.09 \pm 0.05$ | $0.48 \pm 0.00$ | $-0.02 \pm 0.01$ | – |
| **Classical models** | | | | | | |
| Logistic[†] | Summary | 0.68 | 0.17 | 0.56 | 0.05 | 0.27 |
| | Summary+BS+CF+PL | 0.59 | 0.08 | 0.51 | -0.01 | 0.33 |
| | Summary+BS+CF+PL+Text | 0.57 | 0.06 | 0.51 | 0.03 | – |
| Random Forest[†] | Summary | **0.75** | **0.38** | 0.54 | 0.09 | 0.35 |
| | Summary+BS+CF+PL | **0.75** | **0.38** | 0.53 | 0.01 | 0.40 |
| | Summary+BS+CF+PL+Text | 0.73 | 0.31 | 0.52 | 0.03 | – |
| XGBoost[†] | Summary | 0.68 | 0.36 | 0.55 | 0.10 | 0.34 |
| | Summary+BS+CF+PL | 0.63 | 0.27 | 0.53 | 0.06 | 0.42 |
| | Summary+BS+CF+PL+Text | 0.61 | 0.24 | 0.53 | 0.05 | – |

## 5 CONTAMINATION

Since the annual reports in EDINET-Bench are publicly available online, there is a potential risk of test set contamination (Oren et al., 2024; Golchin & Surdeanu, 2024). Several methods have been proposed to detect test set contamination, including prompting models to reproduce verbatim examples from the test set (Golchin & Surdeanu, 2024) and perplexity-based approaches (Mattern et al., 2023; Chan et al., 2025). However, generating verbatim examples is not suitable for assessing whether a model has memorized the correspondence between reports and their labels, and perplexity

Table 7: Feature importance of logistic regression on accounting fraud detection, sorted by absolute value of feature importance scores.

| Feature | Importance | Abs Importance |
|---|---|---|
| Comprehensive Income (CurrentYear) | -1.165 | 1.165 |
| Total Assets (Prior4Year) | -1.118 | 1.118 |
| Total Assets (Prior3Year) | -1.073 | 1.073 |
| Comprehensive Income (Prior3Year) | 1.066 | 1.066 |
| $\vdots$ | $\vdots$ | $\vdots$ |
| Price-to-Earnings Ratio (Prior2Year) | 0.019 | 0.019 |
| Price-to-Earnings Ratio (Prior4Year) | -0.014 | 0.014 |
| Return on Equity (Prior1Year) | -0.008 | 0.008 |
| Price-to-Earnings Ratio (Prior1Year) | 0.001 | 0.001 |

Table 8: Performance of each model on company name prediction. BS+CF+PL+Summary is used as input. The comparison between the ground truth company names and the predicted names is conducted using GPT-4o as a judge to account for inconsistencies in name representations.

| | Claude 3.5 Haiku | Claude 3.5 Sonnet | Claude 3.7 Sonnet | DeepSeek-V3 | DeepSeek-R1 | GPT-4o | o4-mini |
|---|---|---|---|---|---|---|---|
| **Acc.** | 0.045 | 0.050 | 0.045 | 0.000 | 0.005 | 0.000 | 0.005 |

cannot be computed for closed-source models. To evaluate the potential impact of contamination, we instead focus on company name prediction and year-wise performance analysis.

**Company name prediction.** To confirm whether the models already knew the content of the securities reports used in the evaluation dataset, we measure the performance of a task where the model is given tabular data (BS, CF, PL, Summary) from the test split of the accounting fraud detection dataset and asked to predict the company names. The prompt is shown in Figure 7. For evaluation, we accepte minor variations in company names, such as the inclusion or omission of terms like 'Co., Ltd.", using the LLM-as-a-judge approach (Zheng et al., 2023). Table 8 shows the result. For all models, the accuracy is 0.05 or lower, indicating the difficulty for models to associate tabular data with company names.

**Year-wise performance analysis.** To check for potential test set contamination, we analyzed performance by fiscal year on the accounting fraud detection task. Since models may have seen older data during training due to their knowledge cutoff dates, contamination would manifest as higher performance on older years compared to recent years. We split the test set by fiscal start year and plotted yearly performance in Figure 4. The results show no declining trend for recent years, indicating that test set contamination is unlikely to be affecting our evaluation.

Although the above results suggest that contamination has only a limited impact on the experiments, we cannot conclude that the dataset is entirely free of contamination. Achieving a fully contamination-free evaluation would require using annual reports released after the model's cutoff date. This is possible with our toolkit, which iteratively updates datasets with the latest annual reports.

## 6 CONCLUSION

In this work, we introduced EDINET-Bench, an open-source Japanese financial benchmark designed to evaluate the performance of LLMs on challenging financial tasks including accounting fraud detection, earnings forecasting, and industry prediction. We leveraged Japan's Electronic Disclosure for Investors' NETwork (EDINET) to construct EDINET-Bench and can easily incorporate future reports using our toolkit, edinet2dataset. Our experiments reveal that even state-of-the-art LLMs perform only slightly better than logistic regression in binary classification for fraud detection and earnings forecasting. This suggests that simply providing reports to LLMs in a straightforward setting is insufficient. The results underscore the need for benchmark frameworks that better emulate the environments in which financial professionals operate, incorporating richer scaffolding, realistic

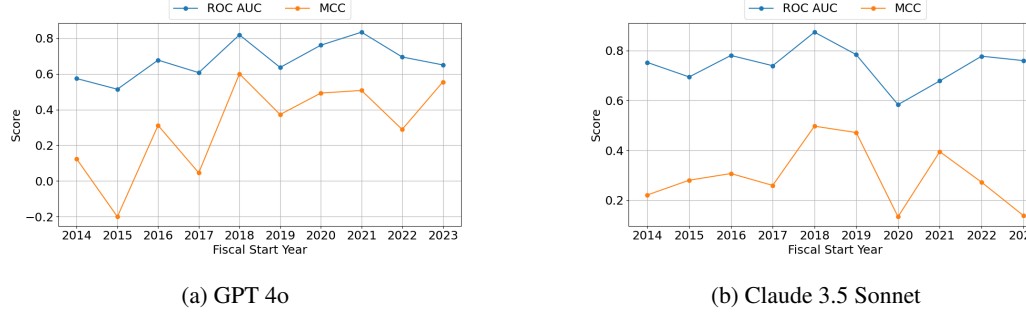

| (a) GPT 4o | (b) Claude 3.5 Sonnet |

Figure 4: Performance per fiscal first year on accounting fraud detection.

simulations, and task-specific reasoning support to enable more effective problem solving. We hope our benchmark and toolkit will accelerate advancements in applying large language models to complex financial analysis tasks.

## FUTURE WORK

Building on this study, several promising directions remain for future exploration.

**Multilingual dataset.** EDINET-Bench is constructed from Japanese annual reports. Extending the benchmark to multiple languages would allow for cross-lingual evaluation of financial document understanding and better reflect the global nature of financial markets. For instance, the U.S. EDGAR system provides functionality analogous to EDINET, suggesting that a comparable dataset construction pipeline could be applied to build a multilingual extension of the benchmark.

**Agentic framework.** In this work, we evaluated LLMs in a simple setting where only a single annual report was provided as input. In practice, however, financial professionals rely on a much broader range of information, such as internal records including receipts and meeting minutes, reports from peer companies, and even real-time economic news. Future research could therefore explore benchmark task designs that enable models to incorporate information beyond the annual report, for example by leveraging agentic capabilities such as web browsing for up-to-date financial news, local file access for company records, or a Python execution environment for quantitative analysis (Wang et al., 2025; Yang et al., 2024). In addition, rather than restricting evaluations to binary classification, benchmarks could adopt *rubric-based evaluation*, where model outputs are assessed according to multiple criteria tailored to each specific question (Starace et al., 2025; Arora et al., 2025).

## ETHICS STATEMENT

We constructed EDINET-Bench using publicly available data from EDINET, which contains no private information, ensuring compliance with privacy regulations. By releasing EDINET-Bench and edinet2dataset, we aim to contribute significantly to the development of LLMs in real-world financial domain applications. However, since the data represents real companies, there is a risk of misuse, such as damaging a company's reputation. This benchmark dataset is intended purely for advancing LLM applications in practical tasks and should never be used to target or harm actual companies. There is also a possibility that this dataset could be misused to make fraud more difficult for LLMs to detect. We strongly discourage any such attempts, especially in real-world applications.

## REPRODUCIBILITY STATEMENT

In this work, we constructed a benchmark dataset and evaluate models using it. To ensure reproducibility and facilitate further research, we publicly release both the dataset and the code used for its construction, allowing others to rebuild and directly use the dataset. For the evaluation, we also release the evaluation code so that the reported results can be fully reproduced.

## ACKNOWLEDGMENTS

We would like to thank Ryuichi Maruyama, Shimpei Fukagai, Shengran Hu, and Robert Tjarko Lange for helpful discussions.

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

## A  INSTRUCTION PROMPT FOR EACH TASK

Figures 3, 5, and 6 show the prompts used in the evaluation of the respective tasks.

---

**Prompt for earnings forecasting**

Please predict whether the "親会社株主に帰属する当期純利益" (Net income attributable to owners of the parent) in the next fiscal year's securities report will increase compared to the current fiscal year, based on the information available in the current year's securities report. - The input is extracted from a Japanese company's securities report. - Some information may be missing and represented as "-" due to parsing errors. - Some attributes are missing and the total does not equal the sum of the parts. Respond in the following format: JSON: "```json { "reasoning": "string", "prob": float (between 0 and 1, probability that the profit will increase), "prediction": int (0: Decrease, 1: Increase) } ```" The current year's extracted securities report is as follows:

---

Figure 5: Prompt for earnings forecasting.

> **Prompt for industry prediciton**
>
> Based on the following financial report, classify the company into one of the Japanese industry categories. You must classify the company into exactly one of these categories: ["食品", "電気・ガス・エネルギー資源", "建設・資材", "素材・化学", "医薬品", "自動車・輸送機", "鉄鋼・非鉄", "機械", "電機・精密", "情報通信・サービスその他", "運輸・物流", "商社・卸売", "小売", "銀行", "金融(除く銀行)", "不動産"] - The input is extracted from a Japanese company's securities report. - Some information may be missing due to parsing errors. Respond in the following format: JSON: "'json { "reasoning": str (reasoning for the classification), "prediction": str (predicted industry category in Japanese) } "' The current year's extracted securities report is as follows:

Figure 6: Prompt for industry prediciton.

> **Prompt for company name prediciton**
>
> Please predict the name of company of the securities report, based on the information available in the current year's securities report. - The input is extracted from a Japanese company's securities report. - Some information may be missing and represented as "-" due to parsing errors. - Some attributes are missing and the total does not equal the sum of the parts. Respond in the following format: JSON: "'json { "reasoning": "string", "prediction": "string" } "' The current year's extracted securities report is as follows:

Figure 7: Prompt for company name prediciton.

## B  ACCOUNTING FRAUD DETECTION

Figure 8 shows the prompt used to assess whether amendment reasons are related to accounting fraud violations. Figure 9 and 10 show examples of model responses.

> **Prompt to detect if the reason of the amendment is related to accounting fraud**
>
> 以下のテキストは訂正有価証券報告書の冒頭部分です。この訂正有価証券報告書が不適切会計、粉飾決算、会計不正に関連しているかどうかを判断してください。
> 特に「提出理由」の部分に着目し、以下のような言葉や表現がある場合は不正会計の可能性が高いと考えられます： - 不適切会計 - 会計不正 - 不正行為 - 粉飾決算 - 会計処理の誤り - 売上の過大計上 - 費用の過少計上 - 資産の過大評価 - 不適切な収益認識 - 監査法人からの指摘 - 社内調査 - 第三者委員会
> 訂正の理由が単純な記載ミスや軽微な修正ではなく、会計上の重大な問題を示している場合は「Yes」と回答し、詳細な説明を提供してください。特に財務諸表（貸借対照表、損益計算書、キャッシュフロー計算書など）の数値に変更が生じた事例に注目してください。会計不正とは関係ない場合や財務諸表に変更が生じていない場合は「No」と回答し、その理由を簡潔に説明してください。
> 以下のJSON形式で回答してください。このJSONは必ず有効なJSON形式である必要があります：
> "'json { "is_accounting_fraud": bool # true or false "explanation": str # 理由を説明する "company_name": str # 会社名 } "'
> 回答は必ずこの形式に一致させてください。 JSONは必ず上記の形式の"'json"'タグで囲んで提供してください。
> 訂正有価証券報告書のテキスト:

Figure 8: Prompt to detect if the reason of the amendment is related to accounting fraud.

## C  EARNINGS FORECASTING

Figure 11 and 12 shows examples of model response.

## D  INDUSTRY PREDICTION

Table 9 shows the list of the 33 and 17 class labels we use for industry prediction. Figure 13 and 14 show examples of model response.

## E  LLM USAGE

We used LLMs to assist with labeling during the dataset construction for the accounting fraud detection task. Additionally, they were employed to help improve the readability of the manuscript during the writing process.

Table 9: Mapping of SICC 33-sector classification to the TOPIX-17 Series.

| No. | 33 Sectors | TOPIX-17 Series |
|---|---|---|
| 1 | Fishery, Agriculture & Forestry | Foods |
| 2 | Foods | |
| 3 | Mining | Energy Resources |
| 4 | Oil and Coal Products | |
| 5 | Construction | Construction & Materials |
| 6 | Metal Products | |
| 7 | Glass and Ceramics Products | |
| 8 | Textiles and Apparels | Raw Materials & Chemicals |
| 9 | Pulp and Paper | |
| 10 | Chemicals | |
| 11 | Pharmaceutical | Pharmaceutical |
| 12 | Rubber Products | Automobiles & Transportation Equipment |
| 13 | Transportation Equipment | |
| 14 | Iron and Steel | Steel & Nonferrous Metals |
| 15 | Nonferrous Metals | |
| 16 | Machinery | Machinery |
| 17 | Electric Appliances | Electric Appliances & Precision Instruments |
| 18 | Precision Instruments | |
| 19 | Other Products | IT & Services, Others |
| 20 | Information & Communication | |
| 21 | Services | |
| 22 | Electric Power and Gas | Electric Power & Gas |
| 23 | Land Transportation | Transportation & Logistics |
| 24 | Marine Transportation | |
| 25 | Air Transportation | |
| 26 | Warehousing and Harbor Transportation | |
| 27 | Wholesale Trade | Commercial & Wholesale Trade |
| 28 | Retail Trade | Retail Trade |
| 29 | Banks | Banks |
| 30 | Securities and Commodities Futures | Financials (ex Banks) |
| 31 | Insurance | |
| 32 | Other Financing Business | |
| 33 | Real Estate | Real Estate |

---

**Accounting fraud detection analysis with Claude 3.7 Sonnet**

After analyzing the securities report, I've identified several unusual patterns that warrant attention: 1. There's a significant jump in parent company net profit (親会社株主に帰属する当期純利益) from 4,431 million yen in the prior year to 7,468 million yen in the current year (+68.5%), despite operating profit (営業利益) decreasing from 8,693 million yen to 5,592 million yen (-35.7%) and ordinary profit (経常利益) decreasing from 8,135 million yen to 6,773 million yen (-16.7%). This unusual profit pattern requires explanation. 2. The source of this profit increase appears to be from investment securities sales (投資有価証券売却益), which jumped from 20 million yen to 4,078 million yen. This one-time gain is being used to mask deteriorating operational performance. 3. Cash flow patterns show inconsistencies: While the company reports increased profits, operating cash flow decreased from 8,400 million yen to 6,042 million yen. Meanwhile, investing cash flow turned positive (from -5,073 million yen to +2,245 million yen), primarily due to proceeds from investment securities sales (投資有価証券の売却による収入) of 5,551 million yen. 4. The company mentions in the risk section that they sold investment securities to secure liquidity due to COVID-19 concerns, but the scale of these sales suggests they may be liquidating assets to artificially boost profits during a downturn. 5. The company delayed publishing their medium-term management plan twice, citing COVID-19 uncertainty. This could be a legitimate reason, but it also provides a convenient excuse to avoid making specific commitments that might be difficult to achieve given their operational challenges. 6. The company has a high debt ratio (31.9% of total assets are borrowings), which creates pressure to maintain profitability metrics. While these observations don't definitively prove fraud, they suggest the company may be engaging in earnings management by using one-time gains from asset sales to mask deteriorating operational performance. This approach is unsustainable and raises concerns about the company's transparency regarding its true financial condition.

Figure 9: Accounting fraud detection analysis with Claude 3.7 Sonnet. Input: BS+CF+PL+Summary+Text, Ground truth: non-fraud.

---

**Accounting fraud detection analysis with GPT-4o**

The securities report appears to be consistent in terms of numerical data and calculations, as stated. However, there are several non-numerical inconsistencies and logical red flags that could suggest potential fraudulent activities or at least warrant further investigation: 1. **Unusual Increase in Parent Company Net Income**: The parent company's net income ('親会社に帰属する当期純利益') increased significantly in the current year (¥7,468 million) compared to the prior year (¥4,431 million), despite a decline in sales ('売上高') and operating profit ('営業利益'). This discrepancy is partially explained by a large gain on the sale of investment securities (¥4,078 million), but the reliance on non-operating income for profit growth could indicate a lack of sustainable operational performance. 2. **Significant Changes in Comprehensive Income**: Comprehensive income ('包括利益') shows a dramatic increase from a near-zero figure in the prior year (¥-60 million) to ¥18,543 million in the current year. This is primarily driven by gains in other comprehensive income items, such as securities valuation differences and retirement benefit adjustments. While these changes may be legitimate, the magnitude of the shift warrants scrutiny. 3. **Cash Flow from Investing Activities**: The cash flow from investing activities turned positive (¥2,245 million) in the current year, primarily due to the sale of investment securities. This reliance on asset sales for cash flow could indicate liquidity issues or an attempt to boost financial metrics artificially. 4. **Unusual Accounting Practices**: The report mentions the use of 'land revaluation' and 'deferred hedge accounting,' which are complex accounting treatments. While these are not inherently fraudulent, they can be used to manipulate financial results if not applied correctly. 5. **Inconsistent Employee Trends**: The number of temporary employees ('平均臨時雇用人員') has decreased significantly over the years, from 255 in the prior four years to 215 in the current year, despite relatively stable total employee numbers. This could indicate cost-cutting measures or operational downsizing, which are not clearly explained in the report. 6. **Related Party Transactions**: The report mentions significant transactions with related parties, such as subsidiaries and affiliates. While these are disclosed, the terms and conditions of these transactions are not detailed, leaving room for potential manipulation. 7. **Complex Corporate Structure**: The company has a large number of subsidiaries and affiliates, including many overseas entities. This complexity can make it easier to obscure financial irregularities. 8. **Unusual Gains and Losses**: The report includes significant one-time gains (e.g., investment securities sales) and losses (e.g., asset impairments). While these may be legitimate, the timing and magnitude of these items could be used to smooth earnings or meet financial targets. 9. **Lack of Transparency in Certain Areas**: Some sections of the report, such as the details of the '2025 Vision' and the use of funds from asset sales, lack sufficient detail to fully assess their impact on the company's financial health. While these issues do not conclusively indicate fraud, they highlight areas of concern that should be investigated further.

Figure 10: Accounting fraud detection analysis with GPT-4o. Input: BS+CF+PL+Summary+Text, Ground truth: non-fraud.

---

**Earnings forecasting analysis with Claude 3.7 Sonnet**

Based on the financial data, I can observe several positive trends that suggest continued profit growth: 1) The company has shown consistent growth in net income over the past 4 years (160,483,000 → 176,451,000 → 217,213,000 → 285,490,000), with a significant 31.4% increase in the most recent year. 2) Sales have been steadily increasing, with a 1.4% growth in the current year. 3) Operating profit increased by 17.6% (327,476,000 → 385,065,000) and ordinary profit increased by 20.0% (330,891,000 → 397,228,000) in the current year. 4) The company has a healthy financial position with increasing net assets (1,974,469,000 → 2,335,993,000) and a strong cash position (2,045,720,000). 5) The company is in the IT solutions industry, which continues to see demand for digital transformation, cloud services, and other technology solutions. 6) The company has been expanding its workforce (605 → 637 employees) to support growth. However, there are some potential concerns: 1) The growth rate of sales has slowed (1.4% vs. previous years' higher growth). 2) Operating cash flow decreased significantly (273,993,000 → 157,958,000). 3) The company mentions potential impacts from COVID-19 on the next fiscal year's performance. 4) There are increasing investments and expenses related to business expansion. Despite these concerns, the overall trend of profitability improvement, the company's strong financial position, and the continued demand in the IT industry suggest that profit growth is likely to continue, albeit possibly at a more moderate pace than the current year's exceptional 31.4% growth.

Figure 11: Earnings forecasting analysis with Claude 3.7 Sonnet. Input: BS+CF+PL+Summary+Text, Ground truth: Increase.

> **Earnings forecasting analysis with GPT-4o**
>
> The company's financial performance has shown consistent growth in key metrics over the past few years. Revenue ('売上高') has increased steadily from 4,964,794,000 yen three years ago to 6,233,729,000 yen in the current year. Operating profit ('営業利益') and ordinary profit ('経常利益') have also grown significantly, with ordinary profit rising from 241,174,000 yen three years ago to 397,228,000 yen in the current year. Net income attributable to owners of the parent ('親会社株主に帰属する当期純利益') has similarly increased from 160,483,000 yen three years ago to 285,490,000 yen in the current year. Additionally, the company has a strong financial position, with increasing net assets ('純資産額') and a high equity ratio ('自己資本比率'). While there are some risks, such as potential economic fluctuations and challenges in securing human resources, the company's strategic focus on IT solutions, including emerging technologies like AI and cloud services, positions it well for continued growth. Based on these factors, it is highly probable that the net income attributable to owners of the parent will increase in the next fiscal year.

Figure 12: Earnings forecasting analysis with GPT-4o. Input: BS+CF+PL+Summary+Text, Ground truth: Increase.

> **Industry prediction analysis with Claude 3.7 Sonnet**
>
> Based on the financial data, this appears to be a financial services company, specifically not a bank. Key indicators include: 1) Very large total assets (3.39 trillion yen) compared to revenue (708.5 billion yen), indicating a capital-intensive financial business. 2) Significant borrowings and lending activities - large short-term borrowings (558.1 billion yen), long-term borrowings (976.1 billion yen), and bond issuances (393.5 billion yen). 3) Negative operating cash flow (-112.1 billion yen) with positive financing cash flow (131.8 billion yen), typical of lending businesses. 4) Low self-capital ratio (12.7%), common in financial institutions. 5) The balance sheet structure shows characteristics of a financial services company with large assets and liabilities related to financing activities. The company appears to be involved in lending or leasing operations based on its financial structure, making it most likely a non-bank financial institution.

Figure 13: Industry prediction analysis with Claude 3.7 Sonnet. Input: BS+CF+PL+Summary, Ground truth: non-bank financial institution.

> **Industry prediction analysis with GPT-4o**
>
> The financial report indicates a company with significant revenue and profit figures, as well as substantial assets and liabilities. The company also has a high level of cash flow activity, including borrowing and bond issuance, which is characteristic of a trading company. Additionally, the presence of investment securities and activities related to dividends and interest income further supports the classification as a trading company. These characteristics align with the '商社・卸売' (trading and wholesale) industry category in Japan.

Figure 14: Industry prediction analysis with GPT-4o. Input: BS+CF+PL+Summary, Ground truth: non-bank financial institution.

