# OpenReview forum: "EDINET-Bench: Evaluating LLMs on Complex Financial Tasks using Japanese Financial Statements"
_ICLR.cc/2026/Conference — ICLR 2026 Poster_

### Official Review · Reviewer_epuV · 2025-10-26

**Soundness:** 3
**Presentation:** 3
**Contribution:** 2
**Rating:** 4
**Confidence:** 5

**Summary:**

This paper introduces EDINET-Bench, an open-source benchmark designed to evaluate Large Language Models (LLMs) on complex financial tasks using Japanese financial statements. The authors highlight the current gap in financial benchmarks, which often focus on simpler tasks, unlike the specialized expertise required in real-world finance.
EDINET-Bench comprises three challenging tasks: Accounting Fraud Detection, Earnings Forecasting and Industry Classification.
The evaluation setup involves testing various closed-source and open-source LLMs in a zero-shot setting, comparing their performance against classical baselines like Logistic Regression and Random Forest. Evaluation metrics include ROC-AUC and MCC for fraud detection and earnings forecasting, and accuracy for industry classification.
Key findings indicate that even advanced LLMs struggle with these tasks, performing only marginally better than logistic regression in binary classification tasks. The results suggest that simply providing raw annual reports to LLMs is insufficient, underscoring the need for benchmark frameworks that better simulate real-world financial environments with richer scaffolding and task-specific reasoning support.

**Strengths:**

1. Authenticity and Challenging Nature of the Dataset: The benchmark is constructed from a large volume of real-world annual reports. The tasks, such as fraud detection and earnings forecasting, are highly challenging and effectively test the capabilities of large language models in complex, expert-level reasoning.
2. Data Quality and Integrity: The risk of test set contamination was convincingly demonstrated to be low through rigorous analysis, such as company name prediction and temporal performance checks. This significantly enhances the credibility and reliability of the evaluation results.
3. Commitment to Reproducibility: The authors clearly commit to open-sourcing the complete dataset (including both the EDINET-Corpus and EDINET-Bench) alongside the data construction toolkit (edinet2dataset). This ensures full transparency and greatly facilitates future research and replication of the reported results.

**Weaknesses:**

1. Limitations in Task Formulation and Evaluation Setup: The benchmark simplifies all three core tasks into binary classification problems. While this formulation facilitates straightforward evaluation, it may not fully capture the continuous and probabilistic nature of professional financial judgment. In practice, for instance, accounting fraud risk assessment often involves estimating a probability or risk score rather than making a binary determination, and earnings forecasting typically focuses on predicting specific numerical values or the magnitude of change.
2. The definition of Expert-level reasoning in the paper is not rigorous enough. It is stated that
"Expert-level reasoning refers to tasks that require integrating information across tables and text, rather than mere extraction or simple computation."
However, on the one hand, multi-table reasoning datasets already exist in the financial domain, such as FinMath[1]. On the other hand, the authors neither provide a quantitative explanation nor cite any references to clarify what constitutes "simple computation," which undermines the clarity and rigor of the definition.
3. Incomprehensive Experimental Evaluation: The selection of LLMs for evaluation, both open-source and closed-source, does not represent the most recent or state-of-the-art models available, even when considering the submission timeline of the paper and the inherent time required for experimentation. This limits the scope of the performance analysis and its relevance to the current frontier of LLM capabilities.

[1]Yilun Zhao, Hongjun Liu, Yitao Long, Rui Zhang, Chen Zhao, and Arman Cohan. 2024. FinanceMATH: Knowledge-Intensive Math Reasoning in Finance Domains. In Proceedings of the 62nd Annual Meeting of the Association for Computational Linguistics (Volume 1: Long Papers), pages 12841–12858, Bangkok, Thailand. Association for Computational Linguistics.

**Questions:**

1. The authors utilized Claude 3.7 Sonnet to generate labels for the accounting fraud detection task. Could you please elaborate on the rationale behind selecting this specific model over other potentially more powerful alternatives available at the time? Furthermore, as this approach relies on a single model for judgment, could you discuss whether and how this might introduce model-specific biases into the dataset, and what impact this could have on the overall label quality and benchmark reliability?
2. How to rigorously demonstrate that the questions in proposed dataset genuinely require expert-level reasoning? Is it because models achieve lower accuracy on this dataset compared to existing benchmarks (a comparison notably absent in the paper)? Is it because the questions necessitate more reasoning steps? Or is it due to a requirement for external, specialized knowledge—a claim that likewise lacks supporting case analysis?

---

> ### Author Response · Authors · 2025-11-21
>
> We sincerely thank the reviewer for the detailed and constructive feedback. We treasure the opportunity to address your concerns and improve our work.
>
> ## Weakness 1: Oversimplified binary task formulation vs. real financial judgment
>
> We agree with this limitation. In this work, we adopted binary classification primarily because publicly available data with accessible labels exist for these settings. In addition, we were also inspired by prior work that has formulated similar problems as binary classification tasks (and which we cite in the paper), so we followed this established convention for consistency and comparability. In practice, subtasks such as estimating a fraud risk score are indeed more realistic, but creating such labels requires domain experts and does not easily scale, which is why we did not include them in this version.
> That said, our results indicate that fully end-to-end classification is challenging. Based on this insight, we plan to work toward constructing more fine-grained subtasks that better reflect real-world financial judgment as a future work.
>
> ## Weakness 2: Unclear and unrigorous definition of expert-level reasoning
>
> We have revised the definition of expert-level reasoning to be more rigorous and now explicitly reference FinanceMATH. FinanceMATH constructs questions by collecting financial terms from Wikipedia and creating problems that require both the correct domain definition and relatively simple calculations (e.g., computing a given year’s GDP from a table listing its components such as consumption and government spending). These tasks require domain knowledge and basic computation, but they are solvable through explicit formulas and surface-level information.
> In contrast, the tasks included in EDINET-Bench, such as fraud detection and earnings forecasting, demand identifying subtle signals embedded in real Japanese securities filings. Solving them requires synthesizing heterogeneous tables and narratives, interpreting financial context, and uncovering patterns that are not directly stated in the text or tables. This level of inference goes beyond multi-table reasoning and simple computation.
>
>
> ## Weakness 3: Missing latest SOTA models in the experimental evaluation
>
> To address this concern, we conducted additional experiments with state-of-the-art models available at the time of the rebuttal: GPT-5 as the strongest closed-source model and Kimi-K2 as a leading open-source model. GPT-5, in particular, demonstrated strong performance on the earnings forecasting task. We have added these results in the revised version.
>
> |  |  | Fraud Detection |  | Earnings Forecasting |  | Industry Prediction |
> | :---- | :---- | :---- | :---- | :---- | :---- | :---- |
> | Model | Input Setup | ROC-AUC | MCC | ROC-AUC | MCC | Acc |
> | GPT-5 | Summary | 0.56 ± 0.00 | 0.00 ± 0.00 | 0.58 ± 0.00 | 0.09 ± 0.02 | 0.21 ± 0.01 |
> |  | Summary+BS+CF+PL | 0.62 ± 0.03 | 0.08 ± 0.03 | 0.62 ± 0.01 | 0.20 ± 0.02 | 0.31 ± 0.01 |
> |  | Summary+BS+CF+PL+Text | 0.67 ± 0.01 | 0.07 ± 0.04 | 0.65 ± 0.01 | 0.21 ± 0.01 |  |
> | Kimi-K2 | Summary | 0.58 ± 0.02 | 0.10 ± 0.08 | 0.46 ± 0.00 | \-0.06 ± 0.00 | 0.14 ± 0.01 |
> |  | Summary+BS+CF+PL | 0.54 ± 0.02  | 0.08 ± 0.03 | 0.53 ± 0.00 | 0.04 ± 0.00 | 0.26 ± 0.01 |
> |  | Summary+BS+CF+PL+Text | 0.59 ± 0.00 | 0.11 ± 0.05 | 0.55 ± 0.02 | 0.07 ± 0.07 |  |
>
>
> ## Question 1: Why Use Claude 3.7 Sonnet, and Does Single-Model Labeling Cause Bias?
>
> For dataset construction, which was carried out around March 2025, we selected Claude 3.7 Sonnet because it was one of the strongest frontier models available and, in our own comparisons, it showed the best Japanese-language performance and instruction-following ability among models such as GPT-4o and Gemini. These properties were important because the labeling process required the model to carefully read amended reports and follow detailed labeling instructions.
> Although the labels are produced by a single model, we believe model-specific bias is limited in this setting. In most amended reports, the description of fraud is explicit, so the model’s task is largely to extract this information rather than infer implicit signals. Because Claude 3.7 has strong instruction adherence, we observed minimal signs of such issues in practice.
>
>
> ## Question 2: How do you prove the tasks truly require expert-level reasoning?
>
> Taking fraud detection as an example, identifying fraudulent behavior requires checking the coherence of trends across multiple tables, and locating subtle inconsistencies within the narrative disclosures. This process requires specialized accounting and economic knowledge and involves multiple reasoning steps that integrate heterogeneous information rather than relying on surface-level cues [1]. This is the sense in which the tasks in EDINET-Bench demand expert-level reasoning.
>
> [1] Satoshi KONDO, et al., 2019. "Using Machine Learning to Detect and Forecast Accounting Fraud,", Research Institute of Economy, Trade and Industry.

---

> > ### Comment · Reviewer_epuV · 2025-11-27
> >
> > Thank you for your response — it addresses part of my concern. I wish to keep my score.

---

### Official Review · Reviewer_rFcH · 2025-10-29

**Soundness:** 4
**Presentation:** 3
**Contribution:** 3
**Rating:** 6
**Confidence:** 5

**Summary:**

EDINET-Bench is an open, reproducible benchmark built from 10 years of Japanese EDINET annual reports to evaluate LLMs on expert-level financial tasks: accounting fraud detection, earnings direction forecasting, and industry classification. The authors release edinet2dataset and the EDINET-Corpus, construct labels , and test frontier closed/open LLMs in zero-shot across multiple input configurations (tables + text). Results show state-of-the-art LLMs struggle and often only match simple classical baselines, highlighting the need for richer scaffolding and agentic settings to better reflect real-world financial analysis.

**Strengths:**

- Real-world, long-context financial documents (tables + text) with open data, code, and tooling for reproducibility.
- Clear task definitions and splits; systematic evaluation across models and input modalities, plus contamination checks.
- Practical insights (e.g., text helps fraud detection but not earnings forecasting) that inform future benchmark and agentic framework design.

**Weaknesses:**

- The benchmark is limited to the Japanese financial market. But I believe the methodology of the benchmark is applicable to other financial markets. Results may be different and interesting in other markets (I guess the industry prediction task results may vary).

- The industry prediction task, while straightforward to evaluate, does not closely reflect real-world use cases. In practice, industry labels are readily available from official sources, so predicting them from financial statements has limited practical value. As such, its significance for assessing LLM capabilities in realistic financial analysis scenarios is questionable.*

- The earnings forecasting task does not incorporate any macroeconomic, industry-wide, or peer company data, despite earnings being strongly influenced by such factors in real-world scenarios. As a result, the task may undervalue the importance of external context and limit the realism of its predictive setting.

**Questions:**

- For each individual sub-task in the paper, how does the work establish novelty? For instance, in the case of fraud detection, there are already existing benchmarks such as:
  Wang, R., Liu, J., Zhao, W., et al. *AuditBench: A Benchmark for Large Language Models in Financial Statement Auditing* [C] // International Workshop on AI for Transportation. Springer, Singapore, 2025: 59-81.

- Why not evaluate stronger tabular models like XGBoost, LightGBM, or CatBoost to provide a stronger baseline in Table 6?

---

> ### Author Response · Authors · 2025-11-21
>
> We thank the reviewer for the thoughtful and detailed feedback. We treasure the opportunity to address your concerns and improve our work.
>
> ## Weakness 1
> > The benchmark is limited to the Japanese financial market. But I believe the methodology of the benchmark is applicable to other financial markets. Results may be different and interesting in other markets.
>
> We agree with this observation. The methodology is directly applicable to other markets, and systems such as the U.S. EDGAR can be used to construct analogous benchmarks. Because model performance can vary with language, reporting standards, and market structure, we expect results to differ across jurisdictions.
>
> ## Weakness 2
> > The industry prediction task, while straightforward to evaluate, does not closely reflect real-world use cases. In practice, industry labels are readily available from official sources, so predicting them from financial statements has limited practical value.
>
> While the task is sometimes regarded as a toy problem, we believe it has meaningful practical applications.
> First, although listed firms are assigned industry categories, these labels tend to remain fixed for long periods, even as companies evolve through M&A or expand into new business areas. For applications such as industry-aware portfolio construction or sector-rotation strategies, it can be more informative to use an industry classification that reflects a firm’s current business rather than its legacy label. (We acknowledge, however, that in our benchmark such cases are treated as misclassifications.)
> Second, institutional investors and financial institutions often employ proprietary industry taxonomies tailored to their investment or risk-management frameworks. Assigning companies to such custom industry classes is costly and labor-intensive. If a model can reliably classify firms into these custom categories based on financial statements, it would provide clear practical value.
> Even aside from these applications, we maintain that the task is useful for evaluating the financial-reasoning abilities of LLMs.
>
> ## Weakness 3
> > The earnings forecasting task does not incorporate any macroeconomic, industry-wide, or peer company data, despite earnings being strongly influenced by such factors in real-world scenarios.
>
> We agree that earnings are influenced by macroeconomic conditions and peer or industry-wide dynamics, which are not captured in our current setup. In future work, we plan to construct a more realistic simulation environment that incorporates macroeconomic indicators, industry context, and peer-company data.
>
> ## Question 1
> > For each individual sub-task in the paper, how does the work establish novelty? For instance, in the case of fraud detection, there are already existing benchmarks such as AuditBench.
>
> Thank you for pointing out AuditBench, which we were not previously aware of. We have reviewed their work carefully. AuditBench focuses on basic discrepancies between transaction data and financial statements, where GPT-4 is used to inject errors and flag them according to accounting standards. The tasks are relatively structured and straightforward, designed to evaluate foundational capabilities in accounting data auditing.  In contrast, our fraud detection task uses real financial statements from Japanese companies. Real-world fraud often involves sophisticated schemes, such as circular transactions, where surface-level calculations appear correct. Detecting such fraud requires reasoning over overall patterns and inconsistencies between company operations and reported financial statements, making it a more challenging and realistic task. Thus, while AuditBench is valuable for assessing basic auditing abilities, our benchmark evaluates expert-level financial reasoning and real-world fraud detection, providing a complementary and more complex setting.
> Furthermore, we have open-sourced both the dataset construction toolkit and the resulting datasets, enabling the research community to develop AI methods for real-world financial tasks.
> We have added this discussion in the revision version.
>
> ## Question 2
> > Why not evaluate stronger tabular models like XGBoost, LightGBM, or CatBoost to provide a stronger baseline in Table 6?
>
> To broaden the coverage of tabular baselines, we evaluated XGBoost. The results showed slightly lower performance than Random Forest for fraud detection, but comparable performance for earnings forecasting and industry prediction.
> We have added these results in the revised version.
>
> |  |  | Fraud Detection |  | Earnings Forecasting |  | Industry Prediction |
> | :---- | :---- | :---- | :---- | :---- | :---- | :---- |
> | Model | Input Setup | ROC-AUC | MCC | ROC-AUC | MCC | Acc |
> | XGBoost | Summary | 0.677 | 0.355 | 0.554 | 0.104 | 0.335 |
> |  | Summary+BS+CF+PL | 0.628 | 0.272 | 0.529 | 0.055 | 0.415 |
> |  | Summary+BS+CF+PL+Text | 0.608 | 0.237 | 0.526 | 0.049 |  |

---

### Official Review · Reviewer_rxpy · 2025-10-31

**Soundness:** 3
**Presentation:** 3
**Contribution:** 3
**Rating:** 6
**Confidence:** 3

**Summary:**

The paper presents EDINET-Bench, an open-source benchmark built from ~10 years of Japanese securities (annual) reports from the FSA’s EDINET system to evaluate LLMs on three expert-level financial tasks: accounting fraud detection, next-year earnings direction forecasting, and industry prediction. The authors release a tool (edinet2dataset) to harvest/parse EDINET TSV/PDFs and construct both a large corpus and three task datasets. Zero-shot evaluations of several closed/open LLMs versus classical baselines show that LLMs perform only marginally better than logistic regression and are outperformed by random forests on fraud detection; earnings forecasting is hard for all models; industry prediction is relatively easier from tables+summary. The paper argues that “just feed the report” is insufficient and calls for richer, more realistic scaffolding.

**Strengths:**

Clear gap & relevance. Financial NLP benchmarks often emphasize QA/extraction (e.g., FinQA/ConvFinQA/FinanceBench; largely English), whereas this work targets end-to-end expert reasoning over full reports in Japanese—a meaningful and under-served setting.

Nontrivial tasks and inputs. Whole-report inputs (tables + text) and tasks like fraud detection are practically impactful and distinct from prior QA settings and multimodal finance QA (e.g., FAMMA).

Open tooling & dataset construction details. The pipeline (EDINET API + TSV parsing via Polars), corpus statistics, class balances, and prompts are described with reasonable transparency.

**Weaknesses:**

Fraud labels rely on LLM-assisted screening of amended reports. Although later manually checked, the pipeline first classifies “amended report reasons” with Claude and then claims <5% label errors. This creates potential circularity (LLM both builds and is evaluated on the dataset domain) and possible systematic biases in what counts as fraud (e.g., wording styles of amendments). A larger human validation and inter-rater agreement would strengthen validity.

Definition of “fraud” is ambiguous. Amendments can be due to material misstatements or benign issues (e.g., shareholder counts). The paper filters for fraud phrases but the ground-truth notion of fraud vs. error could be better anchored (e.g., to JICPA/FSA definitions or enforcement actions). See classic fraud detection literature (e.g., Beneish M-Score) for framing and baselines.

Evaluation setting is narrow. The main headline is that “LLMs struggle in zero-shot long-input.” But practitioner pipelines typically use retrieval over filings plus structured features (ratios, trends) and programmatic reasoning. Without strong RAG/agentic or tool-use baselines (tables→ratios→rules, fraud heuristics, or Beneish-style features), the conclusion may over-generalize. (The paper suggests this as future work, but including such baselines would be very informative.)

**Questions:**

Label audit: How many fraud-flagged amendments did two independent human annotators review? What was Cohen’s κ? Can you release a gold subset with multi-rater consensus?

Negative controls: Did you try Beneish-M-Score or other ratio-based baselines (or hybrid RF + text signals) for fraud detection as stronger non-LLM baselines?

Tool-use baselines: Could you include RAG over earlier filings + programmatic ratio extraction (few carefully designed tools/skills) to test the “richer scaffolding” claim within this paper?

---

> ### Author Response · Authors · 2025-11-21
>
> Thank you for your detailed and constructive review. We appreciate your recognition of the novelty of targeting end-to-end expert reasoning over full Japanese reports, and the transparency of our data construction pipeline. We treasure the opportunity to address your concerns and improve our work.
>
> ## Weakness 1
> > Fraud labels rely on LLM-assisted screening of amended reports. Although later manually checked, the pipeline first classifies “amended report reasons” with Claude and then claims <5% label errors. This creates potential circularity (LLM both builds and is evaluated on the dataset domain) and possible systematic biases in what counts as fraud (e.g., wording styles of amendments). A larger human validation and inter-rater agreement would strengthen validity.
>
> We acknowledge the potential risks of using LLM-assisted screening. However, in our pipeline, the model does not infer fraud from the full financial report or from features used in the benchmark tasks. Instead, it classifies the textual “amended reasons” explicitly written in the amended reports. Therefore, the opportunity for model-induced bias or circularity is limited.
> We agree that expanding human validation and reporting inter-rater agreement would further strengthen robustness, and we would like to incorporate this in future iterations of the dataset.
>
> ## Weakness 2
> > Definition of “fraud” is ambiguous. Amendments can be due to material misstatements or benign issues (e.g., shareholder counts). The paper filters for fraud phrases but the ground-truth notion of fraud vs. error could be better anchored (e.g., to JICPA/FSA definitions or enforcement actions).
>
> We have actually reviewed all cases listed in the FSA’s administrative penalty (surcharge) reports, and confirmed that the vast majority of these enforcement cases are included in our dataset. While our current labeling relies on amendment text and fraud-related phrases, we agree that anchoring the definition more explicitly to JICPA/FSA criteria would further clarify the distinction between fraud-related amendments and benign corrections. We plan to incorporate such formal definitions in future iterations.
>
> ## Weakness 3 and Question 3
> > Evaluation setting is narrow. The main headline is that “LLMs struggle in zero-shot long-input.” But practitioner pipelines typically use retrieval over filings plus structured features (ratios, trends) and programmatic reasoning. Without strong RAG/agentic or tool-use baselines (tables→ratios→rules, fraud heuristics, or Beneish-style features), the conclusion may over-generalize.
>
> We agree with the concern. In the rebuttal, we added a simple baseline that incorporates the Japanese accounting standards into the prompt. When added to GPT-5, this information led to a small but consistent increase in ROC-AUC compared to the no-standards setting. As a next step, we plan to evaluate more realistic and complex pipelines, including retrieval over filings and ratio- or rule-based features.
>
> |  |  | Fraud Detection |  |
> | :---- | :---- | :---- | :---- |
> | Model | Input Setup | ROC-AUC | MCC |
> | GPT-5 | Summary | 0.56 ± 0.00 | 0.00 ± 0.00 |
> |  | Summary+BS+CF+PL | 0.62 ± 0.03 | 0.08 ± 0.03 |
> |  | Summary+BS+CF+PL+Text | 0.67 ± 0.01 | 0.07 ± 0.04 |
> | GPT-5 with Standards | Summary | 0.58 ± 0.01 | 0.00 ± 0.00 |
> |  | Summary+BS+CF+PL | 0.63 ± 0.04 | 0.07 ± 0.01 |
> |  | Summary+BS+CF+PL+Text | 0.68 ± 0.01 | 0.11 ± 0.01 |
>
> ## Question 1
> > Label audit: How many fraud-flagged amendments did two independent human annotators review? What was Cohen’s κ? Can you release a gold subset with multi-rater consensus?
>
> In our current setup, a single annotator manually reviewed each fraud-flagged amendment by checking the original amendment PDF.
> Therefore, Cohen’s κ cannot be computed since multiple annotators were not involved in this stage.
> We acknowledge this as a limitation and plan to release a gold subset with multi-rater consensus in a future version of the dataset.
>
> ## Question 2
> > Negative controls: Did you try Beneish-M-Score or other ratio-based baselines (or hybrid RF + text signals) for fraud detection as stronger non-LLM baselines?
>
> Although we did not explicitly test Beneish-style ratios, our logistic regression and random forest baselines were trained on individual financial statement items, allowing the models to implicitly capture ratio-based relationships among variables.
>
> In addition, we note that the original Beneish M-Score coefficients were estimated using U.S. firms, and prior work has shown that its parameters are not directly transferable across jurisdictions without re-estimation. To the best of our knowledge, there is no validated Japanese adaptation of the Beneish model, so applying the U.S. version to Japanese firms would introduce systematic bias. For this reason, we believe including it as a baseline in our setting would be inappropriate.

---

### Official Review · Reviewer_tbHW · 2025-11-01

**Soundness:** 3
**Presentation:** 3
**Contribution:** 3
**Rating:** 6
**Confidence:** 3

**Summary:**

1. The manuscript discusses the EDINET-Bench: an Open-source Japanese financial benchmark evaluating LLMs on accounting fraud detection, earnings forecasting, and industry classification.

2. The benchmark is very in depth and built from 10 years of EDINET annual reports (~40K docs). The manuscript is worked very comprehensive and seems like a good work

3. Tasks demand integration of text and tables  which is resulting into expert-level financial reasoning. The financial model is well comprehensive.

4. Even top LLMs barely outperform logistic regression. The regression is also well connected and shows some good work done.

**Strengths:**

1. This would be a large-scale Japanese financial benchmark requiring expert reasoning. So the model looks like a good work delivered.
2. Covers diverse, realistic financial tasks. The real world scenario based tasks being financial in nature would be good.
3. Includes reproducible toolkit and dataset which have other applications as well

**Weaknesses:**

1. LLMs show limited performance gains over traditional models. SLMs could also be explored in depth for the tasks. Even MCP route with more sophisticated approaches can be taken into consideration

2. Some of the Evaluation limited to zero-shot; no fine-tuning comparisons. Zero shot learnings in some cases can be a good way to go but cannot be relied on all the time.

**Questions:**

1. How would fine-tuned domain-specific small language LLMs perform when compared with zero-shot learning?
2. Could multilingual or cross-market benchmarks improve robustness, that will be good to explore?
3. What scaffolding or reasoning frameworks could help ?

---

> ### Author Response · Authors · 2025-11-21
>
> Thank you for your thoughtful and positive review. We appreciate your recognition of the benchmark’s scale, realism, and contribution to expert-level financial reasoning, as well as the value of our toolkit and dataset. Your constructive suggestions on model comparisons, fine-tuning, and potential extensions are also very helpful. We address each point below.
>
> ## Weakness 1
> > LLMs show limited performance gains over traditional models. SLMs could also be explored in depth for the tasks. Even MCP route with more sophisticated approaches can be taken into consideration
>
> Regarding the suggestion to explore SLMs (small language models), we note that even frontier models showed relatively weak performance on our benchmark. Given this, we did not expect smaller models to outperform or provide substantial additional insight. That said, we did experiment with a fine-tuned version of Llama-3.2 1B as shown in the paper.
>
> Concerning the “MCP route with more sophisticated approaches,” we understand this as referring to more advanced agentic methods involving tool use, such as web-browsing agents, structured retrieval, or multi-step decision-making frameworks. We agree that such approaches are promising directions that may help models overcome some of the reasoning limitations observed in our experiments. However, these methods introduce substantial system-level complexity and are beyond the scope of this initial benchmark release. We plan to explore such agent-based extensions in future work. For related discussion, please also see our response to Question 3.
>
>
>
> ## Weakness 2
> > Some of the Evaluation limited to zero-shot; no fine-tuning comparisons. Zero shot learnings in some cases can be a good way to go but cannot be relied on all the time.
>
> We clarify that our evaluation is not limited to zero-shot. Table 6 includes fine-tuning results for Llama-3.2-1B SFT, showing performance after supervised training on each task.
>
> ## Question 1
> > How would fine-tuned domain-specific small language LLMs perform when compared with zero-shot learning?
>
> In our experiments, Llama-3.2-1B SFT outperforms Llama-3.3-70B (zero-shot) on all tasks, demonstrating the benefit of task-specific fine-tuning.
>
> ## Question 2
> > Could multilingual or cross-market benchmarks improve robustness, that will be good to explore?
>
> We agree that performance can vary across languages and markets, particularly because accounting, legal, and economic structures differ across countries. For this reason, we view multilingual and cross-market extensions as an important future direction for the benchmark. A natural next step is to incorporate U.S. filings such as EDGAR, which would enable systematic cross-market evaluation.
>
> ## Question 3
> > What scaffolding or reasoning frameworks could help ?
>
> One promising direction is to enable tool use, such as web-search retrieval, which would allow models to access external information on laws, accounting standards, and company operations. Additionally, introducing structured, multi-step reasoning (e.g., separating ratio extraction, cross-year comparison, and final decision making) may help models handle the complexity of these tasks more reliably.
>
> Our goal in releasing this dataset is precisely to enable the community to investigate and develop such higher-level systems. While implementing complex agentic pipelines is beyond the scope of this initial work, we hope that our benchmark will serve as a foundation for future research exploring these richer reasoning frameworks.

---

### Author Response · Authors · 2025-12-02
**Final Author Remark**

We thank the area chair for handling our submission and all reviewers for their thoughtful feedback.
We are encouraged that the reviewers recognized the scale, realism, and reproducibility of EDINET-Bench as key strengths. During the rebuttal, we also addressed concerns raised by the reviewers.
- We clarified that the evaluation is not limited to zero-shot. The paper already reports supervised fine-tuning results, and in the rebuttal we added experiments with the latest closed and open SOTA models, GPT-5 and Kimi-K2. We further showed that providing Japanese accounting standards yields small gains for GPT-5, highlighting the value of richer scaffolded inputs.
- For fraud labels, we explained that our pipeline classifies only amendment reasons rather than full reports, limiting potential circularity and reducing model bias.
- We also refined the definition of expert-level financial reasoning to more concretely articulate the benchmark’s novelty.

Thank you again for the constructive and insightful feedback.

---

### Meta-Review · Area_Chair_khqZ · 2025-12-17

**Summary:**

This paper presents EDINET-Bench, an open-source Japanese financial benchmark that evaluates LLMs on various financial tasks, such as accounting fraud detection, earnings forecasting, and industry classification. The benchmark is built on over 10 years of Japanese securities annual reports from the FSA’s EDINET system. The authors also release a tool (i.e., edinet2dataset) to harvest/parse EDINET TSV/PDFs and construct a large corpus and three task datasets, which would be very helpful for the research community. Extensive evaluations are reported and discussed in the paper.

Overall, this paper is well written. Reviewers recognized this work makes a solid contribution to the community by providing a comprehensive benchmark with open-source tools. The data quality is well contolled. The tasks in the benchmark are practical and clearly defined. The paper also provides many useful insights.

**Reviewer Concerns:**

Reviewers also raised some concerns regarding evaluations, potential applicability of the study to other languages or markets, human validation, missing baselines such as tabular models, missing SOTA LLMs for evaluation, etc.

The authors have provided detailed responses with additional results, such as results of new baselines (e.g., XGBoost) and latest LLMs (e.g., GPT-5), and they have also clarified some evaluation settings in the paper. Most of the concerns from reviewers have been addressed.

Remaining concerns that have not been addressed in the rebuttal include: evaluation of complex reasoning frameworks, human validation, formal definition of fraud, incorporation of additional information for the earnings forecasting task, etc. The authors mentioned that these remaining issues will be studied in their future work.

**Reviewer Scores:**

Initially, this paper received mixed ratings (6, 6, 6, 4). Some of the concerns have been addressed, but there are still remaining concerns. I think the ratings may remain unchanged after discussion.

---

### Decision · Program_Chairs · 2026-01-26

Accept (Poster)